# Epigenetic Silencing of Tumor Suppressor miR-124 Directly Supports STAT3 Activation in Cutaneous T-Cell Lymphoma

**DOI:** 10.3390/cells9122692

**Published:** 2020-12-15

**Authors:** Lidia García-Colmenero, Jéssica González, Juan Sandoval, Yolanda Guillén, Angel Diaz-Lagares, Evelyn Andrades, Arnau Iglesias, Lara Nonell, Ramon Maria Pujol, Anna Bigas, Lluís Espinosa, Fernando Gallardo

**Affiliations:** 1Department of Dermatology, Institut Hospital del Mar d’Investigacions Mèdiques (IMIM), Universitat Autònoma Barcelona, Carrer del Dr. Aiguader, 88, 08003 Barcelona, Spain; lidia.garcia.colmenero@gmail.com (L.G.-C.); eandrades@imim.es (E.A.); 93329@parcdesalutmar.cat (R.M.P.); 2Cancer Research Program Hospital del Mar Medical Research Institute (IMIM), 08003 Barcelona, Spain; jgonzalez3@imim.es (J.G.); aiglesias@imim.es (A.I.); 3Stem Cells and Cancer Research Laboratory-IMIM, CIBERONC, 08003 Barcelona, Spain; yguillen@imim.es (Y.G.); abigas@imim.es (A.B.); 4Epigenomics Unit, IIS, La Fe, 46026 Valencia, Spain; epigenomica@iislafe.es; 5Cancer Epigenomics, Translational Medical Oncology (Oncomet), Health Research Institute of Santiago (IDIS), University Clinical Hospital of Santiago (CHUS/SERGAS), CIBERONC, 15706 Santiago de Compostela, Spain; angel.diaz.lagares@sergas.es; 6Microarray Department-IMIM, 08003 Barcelona, Spain; lnonell@imim.es

**Keywords:** cutaneous T-cell lymphoma, mycosis fungoides, Sézary syndrome, STAT3 (signal transducer and activator of transcription-3), JAK/STAT (janus kinase/signal transducer and activator of transcription), miR-124

## Abstract

Increasing evidence supports a potential role for STAT3 as a tumor driver in cutaneous T-cell lymphomas (CTCL). The mechanisms leading to STAT3 activation are not fully understood; however, we recently found that miR-124, a known STAT3 regulator, is robustly silenced in MF tumor-stage and CTCL cells. Objective: We studied here whether deregulation of miR-124 contributes to STAT3 pathway activation in CTCL. Methods: We measured the effect of ectopic mir-124 expression in active phosphorylated STAT3 (p-STAT3) levels and evaluated the transcriptional impact of miR-124-dependent STAT3 pathway regulation by expression microarray analysis. Results: We found that ectopic expression of miR-124 results in massive downregulation of activated STAT3 in different CTCL lines, which resulted in a significant alteration of genetic signatures related with gene transcription and proliferation such as MYC and E2F. Conclusions: Our study highlights the importance of the miR-124/STAT3 axis in CTCL and demonstrates that the STAT3 pathway is regulated through epigenetic mechanisms in these cells. Since deregulated STAT3 signaling has a major impact on CTCL initiation and progression, a better understanding of the molecular basis of the miR-124/STAT3 axis may provide useful information for future personalized therapies.

## 1. Introduction

Recent studies have explored the mutational profile in cutaneous T-cell lymphomas (CTCL), particularly in Mycosis Fungoides (MF) and Sézary Syndrome (SS). Results from these studies indicate that patients with MF or SS have complex, heterogeneous, unspecific, or non-recurrent genomic alterations. Previously known and newly described somatic mutations, copy number variations, and gene fusions involving T-cell activation, apoptosis, cell cycle, DNA repair, or chromatin remodeling may contribute to CTCL development or progression [1,2,3,4,5,6,7,8,9,10]. Activating mutations or copy number gains in JAK/STAT pathway components have become an interesting field for research in CTCL and promising targets for therapy [3,5,10,11].

The JAK/STAT pathway is recognized as one of the major mechanisms by which cytokine receptors transduce intracellular signals to the target gene promoters in the nucleus, providing a mechanism for transcriptional regulation. This signaling pathway influences normal cell survival and growth mechanisms and its deregulation may contribute to oncogenic transformation [12,13,14]. Increasing evidence supports a potential role of STAT3 as a tumor driver in CTCL [3,5,9,10,15,16,17,18,19], and most human SS cells and CTCL cell lines display constitutively activate phosphorylated STAT3 (p-STAT3) with STAT3 inhibition resulting in massive apoptosis [20,21,22]. Recently, it has been demonstrated that mice carrying an activated STAT3 transgene develop a lymphocytic syndrome similar to human SS [23]. Nevertheless, the precise mechanism by which STAT3 activation contributes to CTCL and the upstream elements that regulate the pathway in this particular disease remain poorly understood.

Recently, we found that the promoter region of miR-124, a putative regulator of STAT3 in different types of cancer [24,25,26,27], was robustly methylated in CTCL cell lines (including MF and SS) compared with inflammatory disease (ID) skin samples [28]. Herein, we aim to investigate the functional contribution of miR-124 regulation to STAT3 pathway activation in CTCL.

## 2. Materials and Methods

### 2.1. Study Samples, Cell Lines, and Treatments

Fresh-frozen samples from 12 patients diagnosed with Mycosis Fungoides tumor stage (MFt) and 4 representative human cancer CTCL cell lines (American Type Culture Collection, Manassas, VA, USA) Myla (MF), HuT-78, SeAx (SS), and HH (aggressive non-MF CTCL) were used in the different experiments. Control samples for the different study methods included: frozen samples from cutaneous inflammatory diseases (ID) showing a dense lymphocytic infiltrate (psoriasis, *n* = 9), a validation data set as an independent cohort of paraffin-embedded MFt samples (*n* = 19), ID controls (psoriasis, *n* = 7) as well as healthy skin (normal tissue from healthy donor) as a normalization data set. 

### 2.2. DNA Methylation Status of the miR-124 Promoter in CTCL Cell Lines and Patients

DNA from these cell lines and patients was quantified by Quant-iT Pico Green dsDNA Reagent (Invitrogen, Thermo Fisher Scientific, Inc., Waltham, MA, USA) and the integrity was analyzed in a 1.3% agarose gel. Bisulphite conversion of 600 ng of each sample was performed according to the manufacturer’s recommendation for Illumina Infinium Assay (Illumina, Inc. San Diego, CA, USA). Effective bisulphite conversion was checked for three controls that were converted simultaneously with the samples. Four microliters of bisulphite-converted DNA were used to hybridize on Infinium Human Methylation 450 Bead Chip, following Illumina Infinium HD Methylation protocol. Chip analysis was performed using the Illumina HiScan SQ fluorescent scanner. The intensities of the images are extracted using Genome Studio (2010.3) Methylation module (1.8.5) software from Illumina. Methylation score of each CpG is represented as β-value. In patients, for determining differences in miR-124 DNA promoter methylation, non-parametric Wilcoxon tests were applied. Probes with multiple test-corrected *p*-value below 0.05 and a delta value threshold of 0.2 in absolute value were used for selecting the relevant CpGs. All statistical analyses were performed with the Bioconductor project (v2.3) in the R statistical environment (v.2.8.1).

### 2.3. Expression Analysis of miR-124 by qRT–PCR

Total RNA extraction was obtained using the RNeasy Kit from Qiagen (Hilden, Germany). RNA was quantified using a NanoDrop ND-1000 Spectrophotometer (Axon Instruments, Sunnyvale, CA, USA) and analyzed using the RNA Nano 6000 (Bioanalyzer 2100, Agilent Technologies, Palo Alto, CA, USA) in order to analyze the RNA integrity. Samples included in the study had a quality index RIN (RNA integrity number) ≥6. Complementary DNA retro-transcription was synthesized with the qRT first-strand complementary DNA synthesis kit (Amersham Pharmacia Biotech, Piscataway, NJ, USA) following the manufacturer’s instructions. qRT–PCR in triplicate on the Light Cycler 480 (Roche Molecular Systems Inc., Rotkreuz, Switzerland) was performed using a TaqMan Universal PCR Master Mix kit and the ABI PRISM 7900HT Sequence Detection System 7900HT (TaqMan microRNA assays, Applied Biosystems, Foster City, CA, USA) and finally analyzed with SDS software. Each qRT–PCR product concentration (Ct) value was normalized (threshold cycle) relative to endogenous controls.

Validation data set that included an independent cohort of paraffin-embedded MFt samples (*n* = 19) and ID controls (psoriasis, *n* = 7). The diagnosis of MF samples was established according to the World Health Organization/European Organization for Research and Treatment of Cancer classification [29]. Total RNA was extracted from 10 × 10 μm paraffin embedded tissues from the retrospective patient cohort using the commercial kit (MirVanaTMmiRNA Isolation Kit) following the manufacturer’s protocol (Ambion, Austin, TX, USA). For coding genes, keratin 14 (previously normalized by glyceraldehyde 3-phosphate dehydrogenase) was used to normalize overexpression of miR-124 from epidermal component in healthy skin normalization data set and U48 was used in microRNA quantification. The 2-(ΔΔCt) method was used to determine relative quantitative levels of each gene or microRNAs. In short, ΔCt = Ct sample of interest—Ct control) and compared with control patient samples (reference sample), by the method ΔΔCt (ΔΔCt = ΔCt sample of interest—ΔCt reference sample).

### 2.4. Lentiviral Transfection

Ectopic expression of miR-124 in CTCL cell lines infected with lentivirus encoding for miR-124 was evaluated. CTCL cell lines were transient transfected with a lentiviral vector encoding miR-124 (HEK293T cells, DVR8.2 [packaging], PMD2G [envelope]; Addgene, MA, USA). Twenty-four hours later supernatant was ultracentrifuged and the viral pellet was resuspended in 100 mL of phosphate-buffered saline. Twenty microliters of fresh viral suspension were used per infection. Transfected cells were selected in puromycin containing medium and analyzed by Western blot for STAT3 and STAT3 levels. The impact on STAT3 signaling was evaluated using expression microarray (Agilent Human microarrays v3, ID021827; Agilent Technologies, Santa Clara, CA, USA) to compare transcriptional profiles of control CTCL cells and miR-124 expressing cells (after lentiviral transduction).

### 2.5. Microarray Expression Profiles

To study the transcriptional impact of miR-124 expression in CTCL, microarray expression analysis on control CTCL cells and cells transduced with the lentiviral miR-124 construct was performed. We arbitrarily fixed a minimum fold change of 1.5 and a *p*-value ≤ 0.05 as the threshold cut-off for selection of differentially expressed genes. RNA samples obtained from both miR-124 transfected/non-transfected cell lines were processed according to the manuals GeneChip 3′ IVT Pico Reagent kit (P/N 703,308 Rev. 1) and Expression Wash, Stain and Scan User Manual (P/N 702,731 Rev. 3) by Affymetrix Inc. (Santa Clara, CA, USA). Whole transcriptome analysis was performed by using Affymetrix GeneChip PrimeView Human Gene Expression Array in a GeneChip hybridization oven 640. After quality control of raw data, they were background corrected, quantile-normalized and summarized to a gene-level using the robust multi-chip average (RMA) [30]. Linear Models for Microarray (LIMMA from Bioconductor) [31], a moderated *t*-statistics model, was used for detecting differentially expressed genes between the two conditions. Genes with a *p* < 0.05 and with an absolute fold change (FC) value above 1.5 were selected as significant. Statistical analyses were performed using R (v 3.3.0, R Core Team, Vienna, Austria) with Affymetrix and LIMMA packages. 

### 2.6. Gene Ontology Analysis of Differentially Expressed Genes

Functional profiles were assessed using Ingenuity Pathway Analysis (IPA, http://www.ingenuity.com/, Qiagen, Hilden, Germany). Overrepresentation analysis of Gene Ontology (GO) categories and pathways was used to explore the functions associated with miR-124 regulated genes in CTCL. Gene set enrichment analysis (GSEA) software was run using as input the matrix with the normalized expression values of all genes. Hallmark and curated gene sets were used setting a significance threshold of a *p*-value < 0.05.

### 2.7. Ethical Considerations

The approval for the study was provided by the Comitè Ètic d’Investigació Clínica from l’Insititut Municipal d’Assistència Sanitària (CEIC-PSMAR) and written informed consent was obtained from all patients according to the national and international guidelines (code of ethics, the Declaration of Helsinki Principles) and the legal regulations on data privacy (Law 15/1999 of 13 December on the Protection of Personal Data (Data Protection Act) were considered. All samples stored in the tumor bank (Hospital del Mar in Barcelona, Spain) have the informed consent of patients.

## 3. Results

### 3.1. Silencing of miR-124 as a Consequence of Its Promoter Hypermethylation in Both CTCL Cell Lines and MF Tumor Samples

By the analysis of a 450K DNA methylation microarray data from our group [32] (submitted to GEO), we detected a significant hypermethylation of the miR-124 promoter in the CTCL cell lines HuT78 and HH (Figure 1A). The treatment of several CTCL cell lines with the DNA demethylating agent 5-Aza-2′-deoxycytidine (AZA) significantly increased the expression levels of miR-124 (Figure 1B).

By 450K analysis in MFt samples, we confirmed the presence of highly methylated CpG islands in the proximal miR-124 promoter (Appendix A, Figure 1C). However, miR-124 promoter expression in this MFt cohort was not significantly different when compared with ID samples, but showed a statistical trend if outliers were excluded (Mann–Whitney test, *p*-value = 0.0714, trend *p* < 0.01; *t* test, *p*-value = 0.0449) (Figure 1D). Expression of miR-124 by qRT-PCR in an independent MF cohort of paraffin skin samples regarding ID disclosed no significant results (data not depicted).

### 3.2. Hypermethylation of the miR-124 Promoter in CTCL Cells Leads to miR-124 Silencing and Increased STAT3 Levels

CTCL cell lines contain variable levels of activated p-STAT3 (Figure 2) [21,22]. We tested the possibility that miR-124 levels contribute to the regulation of STAT3 signaling in CTCL by transducing a lentiviral vector encoding miR-124. Ectopic miR-124 expression significantly reduced p-STAT3 levels in CTCL cells, as determined by Western blot analysis (see lanes 5 and 6, Figure 2). Comparable effects were detected in cells transduced with the miR-124 construct together with control vector, with the exception of HH cells, which contain the highest levels of p-STAT3 (lanes 3 and 4, Figure 2), likely due to dilution of miR-124 expression vector. As a negative control, increased miR-124 expression did not affect levels of the Notch ligand Jagged1 (Figure 2) that we previously found associated with human CTCL and regulated by miR-200c in these cells.

### 3.3. Ectopic miR-124 Expression in CTCL Cells Imposes an Altered Gene Expression Profile

We performed microarray expression analysis of control (non-transduced) and lentiviral miR-124-transduced CTCL cells (data submitted to GEO: GSE130809). Principal component analysis (PCA) based on total gene expression showed that control (p-STAT3 positive) and miR-124-transduced (*p*-STAT3 negative) CTCL cells clustered separately (Figure 3A). We identified 1858 differentially expressed genes (DEG) in miR-124-transduced cells (compared to control cells) with *p* < 0.05 and |FC| > 1.5 (data deposited in GEO, GSE130809). Ingenuity Pathway Analysis revealed that DEG in miR-124-transduced cells grouped into specific functional categories including MYC, MAPK/KRAS, p53, NF-κB or the JAK/STAT pathways (not depicted). Interestingly, genes downregulated in the miR-124-transduced condition include genes that have been associated with T-cell function and STAT3 regulation such as CCR8 [33], RPTOR [34,35], EED [36], or CYBB [37]. In contrast, genes upregulated in miR-124 expressing cells were mainly associated with inflammation and immune response such as IFNL1 [38], CYP1A1 [39], IL26 [40], CXCL9 [41], and MYD88 [42] (Figure 3B). Additionally, GSEA showed that MYC and E2F target genes were downregulated in miR-124-transduced cells, and these cells display expression dynamics comparable to that observed after miR-21 inhibition in glioma cells [43,44] (Figure 3C).

### 3.4. CTCL Cells Carrying STAT3 Activation Are Highly Sensitive to Pharmacologic JAK/STAT Inhibition

Finally, we performed a dose–response assay in CTCL cells using the JAK/STAT pathway inhibitor Ruxolitinib. We found that low idoses of the inhibitor (1 μM) completely precluded JAK/STAT activation in these cells as indicated by the absence of phosphorylated STAT3, independent of the basal amount of the protein (Figure 4A). Importantly, analysis of cell numbers after 48 h of Ruxolitinib treatment indicated a superior sensitivity of HH and SeAx cells, which contain higher amounts of p-STAT3 under basal conditions (Figure 4B).

Together, our results indicate that epigenetic downregulation of miR-124 is responsible for supporting high p-STAT3 levels in CTCL cells. In addition, our data suggest that p-STAT3 levels in these cells are predictive of their sensitivity to specific JAK/STAT inhibitors, thus highlighting a putative role of the miR-124/STAT3 axis as a therapeutic biomarker in CTCL.

## 4. Discussion

Epigenetic mechanisms leading to miR-124 dysregulation have been previously associated with increased STAT3 activity in different tumors [24,25,26,27], and abrogation of miR-124 in CTCL cells secondary to hypermethylation in the miR-124 promoter has been demonstrated [28]. We have now shown that miR-124 is transcriptionally repressed by promoter methylation in CTCL cells contributing to sustain STAT3 signaling. miR-124 re-expression in these cells modifies important transcriptional programs including those regulating proliferation, chemotaxis, and T-cell function.

STAT3 is one of the nodes connecting transcriptional programs that are altered in CTCL [15,45] with a major impact on cell survival. Different mechanisms have been reported to deregulate STAT3 in CTCL. Genomic alterations including activating mutations, copy number alterations, or gene fusions involving JAK1, JAK2, JAK3, STAT3, and STAT5B genes [46]. In addition, external environmental agents can activate oncogenic STAT3/STAT5 signaling in malignant T cells leading to the release of IL-2 and other cytokines that eventually result in tumor progression [19,23,24,47]. Upregulation of a chemokine receptor CCR6 and its ligand CCL20 has also been shown to promote STAT3 pathway activation in advanced CTCL [48]. Similarly to other hematological neoplasms, this pathway could also be a potential therapeutic target in CTCL [5,49].

Several genes and pathways that are relevant in cancer initiation and progression have been identified in our study as dysregulated by the miR-124/STAT3 axis in CTCL. IFNL1 gene is a recently discovered IFN family important in epithelial innate immune defense and known to be regulated throughout NF-κB, which resulted in a significant elevation of secreted IFNL1 [50]. Our study has also detected alterations in the expression of EED (a polycomb repressive complex 2 gene) and IL26 (a member of the IL10 superfamily which has a role in host innate immune response and is overexpressed in T-cells with proinflammatory properties); both genes differentially expressed in CTCL and benign skin biopsies [36]. Interestingly, IL26 has been shown to promote proliferation and survival by modulating STAT1/STAT3 signaling [51].

CCR8 was significantly downregulated in CTCL through miR-124/STAT3. In cutaneous anaplastic large cell lymphoma, CCR8 downregulation has been postulated to contribute to the low tendency to disseminate to extracutaneous sites [52]. However, a relationship between CCR8 expression and clinical aggressiveness in MF needs to be demonstrated. A regulatory protein of mTOR1 (RPTOR) was upregulated when miR-124 was silenced in CTCL cell lines. This gene has an important role in tumorigenesis and early T-cell development, and could be of interest as future therapeutic target [34]. CXCL9, which has been associated with loss of epidermotropism in advanced CTCL, was significantly downregulated. BCR (breakpoint cluster region) or ATP1B2 (ATPase transporting beta 2) are also important genes commonly involved in CTCL genomic alterations deregulated throughout the miR124/STAT3 axis.

Ceacam1L promotes proliferation of glioblastoma cells by activating the STAT3 function [53], or Syndecan that suppresses T-cell activation in Sézary cells [54]. Conversely, the STAT3 and NF-κB pathways can be disrupted upon Syndecan ablation in breast cancer via Notch signaling [55]. Interestingly, we have observed that miR-124-mediated downregulation of DOK-1 and IL-7. DOK-1 usually decreases both JAK-2 and STAT-3/5 phosphorylation in lymphoma cells. Consistently, silencing of the DOK-1 gene resulted in the rescue of MAP kinases and JAKs activities in CD45 positive T-cells. Downregulation of IL-7 may explain the cytokine-independent activation of STAT3 and STAT5 in CTCL at advanced stages driven exclusively by constitutively active JAK1 and JAK3 kinases.

MYD88 mutations have been reported to be pathogenetically relevant in some cutaneous diffuse large B-cell lymphomas, leg type, and curiously seem to be mutually exclusive from other cancer-promoting mutations that activate the NF-κB pathway including BRAF, PIK3, or STAT3. Our observation may suggest a possible role for MYD88 in CTCL as a consequence of epigenetic regulation [56]. We observed that C-MYC could be another interesting target for the miR-124/STAT3 signaling since it has been commonly deregulated through recurrent genomic disturbances in CTCL [57].

Finally, the observation of other different gene deregulation as a possible target for the miR124/STAT3 axis may be relevant as new markers. In this sense, epigenetic deregulation of CYP1A1 could be of interest since genetic polymorphisms in CYP1A1 gene have been related to a higher risk of non-Hodgkin lymphoma [58]. The ATP1B2 gene, located in 17p13 (close to TP53 locus) and ODF1, located 8p22 (close to C-MYC locus), are frequently rearranged among various types of lymphoma including CTCL. These findings may suggest that epigenetic alterations on these genes may play a role in clonal heterogeneity [59].

In conclusion, epigenetic mechanisms are important events regulating the STAT3 pathway in CTCL. Altogether, our study indicates that the miR-124/STAT3 axis plays a relevant mechanism in CTCL pathogenesis thorough the regulation of a whole transcriptional signature particularly involving the MYC pathway, which is an essential tumor driver in multiple systems, among other factors (Syndecan, TFAP2C, RPTOR DOK-1, IL-7, IFI35, CXCL9, or BCR) that may constitute potential therapeutic targets in CTCL. However, different mechanisms other than promoter methylation may contribute to miR-124 regulation in primary CTCL, including signals derived from non-transformed accompanying cells such as reactive lymphocytes or keratinocytes. Since this additional regulation may be particularly relevant in the incipient lesions, we will analyze in the near future early MF samples, and determine the role of mir124 and its regulation in the development and progression in MF. It would also be of interest to investigate the possible impact of miR-124 modulation in other tumor driver pathways in addition to STAT3.

## Figures and Tables

**Figure 1 cells-09-02692-f001:**
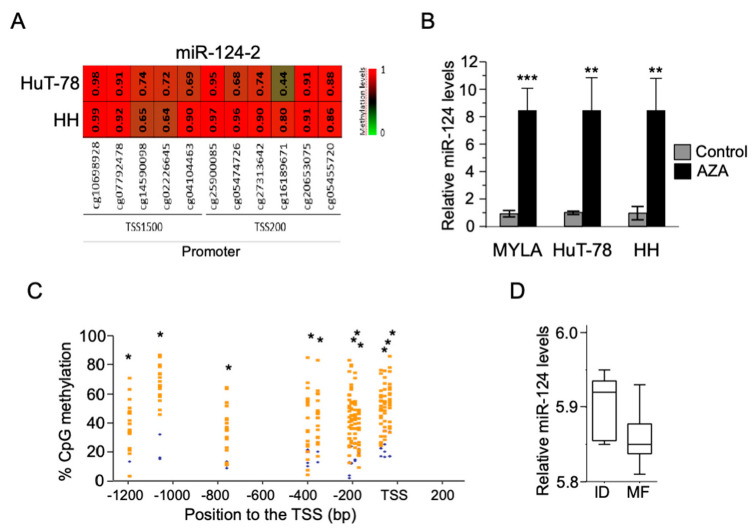
Hypermethylation and silencing of miR-124 in CTCL cell lines and MF tumor samples (**A**) DNA methylation levels of consecutive promoter CpGs from miR-124 in CTCL cell lines analyzed by the 450K DNA methylation microarray. Gradient color bar reference means Green (0, minimum) unmethylated, Red (1, maximum) hypermethylated; (**B**) restored expression of DNA methylated miR-124 in CTCL cell lines after treatment with the DNA demethylating agent 5-aza-2′-deoxycytidine. qRT–PCR product concentration (Ct) value for U6 was used to normalize overexpression of miR-124 in CTCL cell lines by the method ΔΔCt (ΔΔCt = ΔCt sample of interest—ΔCt reference sample) *** *p* < 0.001, ** *p* < 0.01; (**C**) Differentially methylated CpGs in miR-124 were identified in the comparison between MFt (orange) and inflammatory dermatoses (ID) (blue). * *p* < 0.05; (**D**) downregulation of miR-124 expression associated with its promoter hypermethylation in MFt samples. miR-124 expression showed lower levels in MFt samples respect to ID controls. Mann Whitney test, *p*-value = 0.0714, trend *p* < 0.01; *t* test, *p*-value = 0.0449 (statistical significance was reached when outliers were not considered).

**Figure 2 cells-09-02692-f002:**
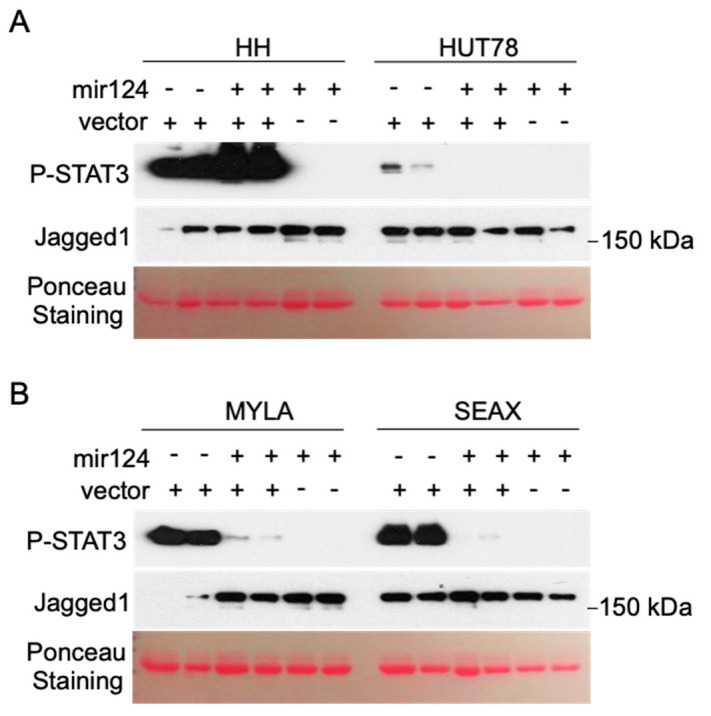
Ectopic expression of miR-124 precludes JAK/STAT3 signaling. Western blot analysis of *p*-STAT3 in HH, HUT78 (A), MYLA and SEAX (B) CTCL cell lines transduced with control vector, control vector plus lentiviral miR-124 expression vector, or miR-124 vector alone. Two biological replicates of each condition are included. Levels of the Notch1 ligand Jagged1 are shown as negative control of miR-124-dependent regulation. It is worth mentioning that the combination of control virus and miR-124 in HH (A) did not produce any effect on p-STAT3 likely due to dilution of the miR-124 lentivirus.

**Figure 3 cells-09-02692-f003:**
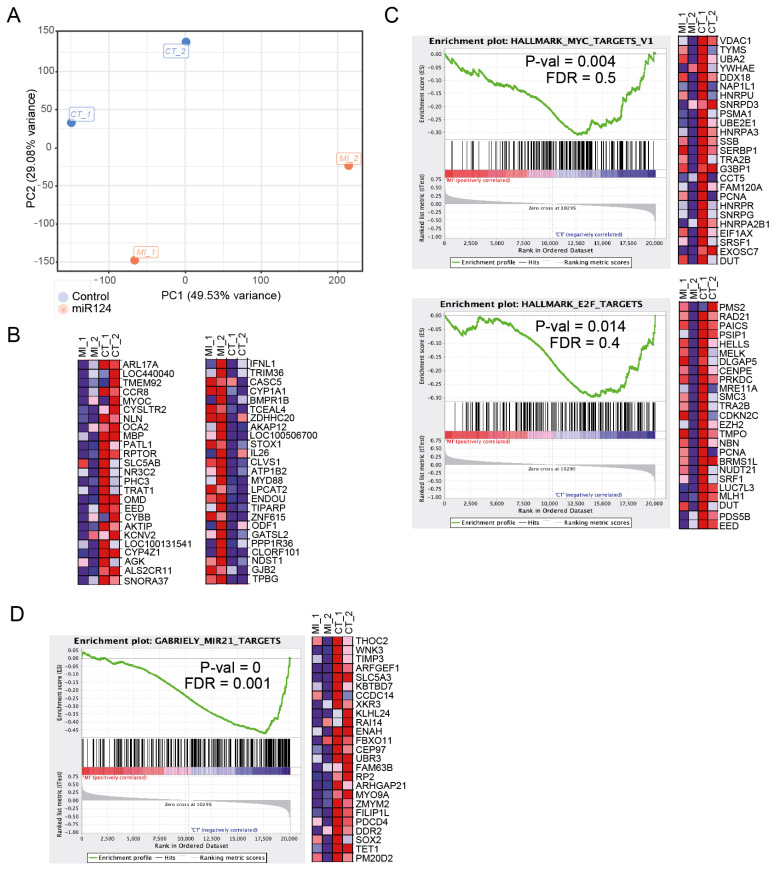
Transcriptional analysis of control and miR-124 transduced CTCL. (**A**) Principal Component Analysis of gene expression in control (non-transduced) CTCL cell lines (CT) versus miR-124-transduced (MI) CTCL cell lines; (**B**) heatmap showing the top differentially expressed genes (upregulated, in red or downregulated, in blue) comparing miR-124-transduced with control non-transduced cells. (**C**,**D**) Gene Set Enrichment Analysis of genes downregulated following miR-124 expression identified a significant association with genes in the MYC and E2F pathways (**C**) and also with genes downregulated by ectopic miR-21 expression (**D**). FDR, false discovery rate.

**Figure 4 cells-09-02692-f004:**
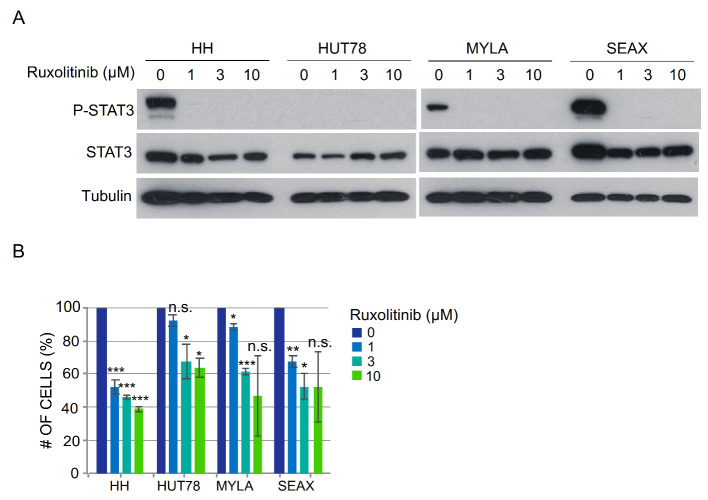
Sensitivity of CTCL cells to pharmacologic JAK/STAT inhibition. Dose–response assay in CTCL cells using the JAK/STAT pathway inhibitor Ruxolitinib. (**A**) low inhibitor doses completely precluded JAK/STAT activation in CTCL as indicated by the absence of p-STAT3; (**B**) Analysis of cell number after 48 h of Ruxolitinib treatment indicated highest sensitivity of HH and SeAx cells, which contain bigger amounts of STAT3 and p-STAT3 under basal conditions. The graph represents the average value and standard deviation of two independent experiments. Statistical significance was calculated by an unpaired *t*-test. *** *p* < 0.001, ** *p* < 0.01, * *p* < 0.05, n.s. = non-significant.

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
