# Peer review of "Epigenetic Silencing of Tumor Suppressor miR-124 Directly Supports STAT3 Activation in Cutaneous T-Cell Lymphoma"

_cells, 2020, doi:10.3390/cells9122692_

Round 1

Reviewer 1 Report

Review Cells manuscript 1006819

In the manuscript entitled “Epigenetic silencing of tumor suppressor miR-124 directly supports STAT3 activation in Cutaneous T-cell Lymphoma” the authors investigated the regulatory role of miR-124 on the STAT3 pathway in CTCL. The authors previously found, that the promotor region of miR-124 is methylated in CTCL cell lines. In the present study the authors found, that miR-124 hypermethylation inhibits the expression of miR-124 and increases the expression of p-STAT3. Conversely, ectopic expression of miR-124, induced by cell transfection, inhibits the protein expression of p-STAT3. In addition, gene expression array analysis revealed an altered expression of genes associated with STAT3 regulation and T-cell function in miR-124 transduced cells. Finally, the authors conclude that the epigenetic downregulation of miR-124 in CTCL may support high p-STAT3 expression and that the miR-124/STAT3 pathway may serve as a therapeutic biomarker in CTCL.

Comments to the authors:

This is an interesting study, that underline the importance of epigenetic changes in the pathogenesis of CTCL. The study is well performed and the manuscript is well-written. However, the manuscript needs some clarifications:

  1. Figure 1A: Did you investigate the hypermethylation level in the Myla cell line?
  2. Figure 1C indicate that the violet line illustrates control samples whereas the corresponding figure legend indicates that it illustrates tMF. Similarly, the indication of the blue line in the figure differs from its legend. Please explain or revise the figure.
  3. Figure 1D: Please correct DI to ID on the x-axis. Why was U48 chosen as normalizer?
  4. The authors found, that ectopic miR-124 expression reduced the level of p-STAT3. Did the authors also investigate the effect on cell proliferation?
  5. Please add a short description of the study limitations.

Author Response

Reviewer 1

Figure 1A: Did you investigate the hypermethylation level in the Myla cell line?

Answer: In our 450k methylation array database we only have results from miRNA promoter methylation on Hut78 and HH, as representative CTCL cell lines. Nevertheless, using the demethylating agent AZA in MyLa cells (Figure 1B) we observed the same response in mir124 expression as in Hut78 and HH cell lines.

Figure 1C indicate that the violet line illustrates control samples whereas the corresponding figure legend indicates that it illustrates tMF. Similarly, the indication of the blue line in the figure differs from its legend. Please explain or revise the figure.

Answer: We have corrected figure 1C.

Figure 1D: Please correct DI to ID on the x-axis. Why was U48 chosen as normalizer?

Answer: We have corrected this. About the use of U48 as normalizer, we apologize for the mistake, and we have removed the sentence “RT–PCR product concentration (Ct) value for U48 was used to normalize overexpression of miR-124 in MF and ID samples compared to healthy skin by the method ΔΔCt (ΔΔCt = ΔCt sample of interest—ΔCt reference sample” from the figure legend. In fact, an independent validation in paraffin-embedded skin samples MF and ID cohorts of (data not depicted), we observed less variability using U6 as normalizer than U48. According to the reviewer comment this error has been revised and modified.

The authors found, that ectopic miR-124 expression reduced the level of p-STAT3. Did the authors also investigate the effect on cell proliferation?

Answer: We haven’t look at proliferation since ectopic miR-124 primarily induced cell death that was observed in the microscopy analysis. Thus, we had a very limited number of cells for our studies.

Please add a short description of the study limitations.

Answer: We have included this in the discussion section of the text: However, different mechanisms other than promoter methylation may contribute to miR-124 regulation in primary CTCL, including signals derived from non-transformed accompanying cells such as reactive lymphocytes or keratinocytes. Since this additional regulation may be particularly relevant in the incipient lesions, we will analyze in the near future early MF samples, and determine the role of mir124 and its regulation in the development and progression in MF. It would be also of interest to investigate the possible impact of miR-124 modulation in other tumor driver pathways in addition to STAT3.”

Reviewer 2 Report

In the manuscript by García-Colmenero et al., submitted to Cells, the authors describe the significance of miR-124 in CTCL, and in particular its role in controlling the activity of STAT3 (phospho-STAT3).

Overall, the manuscript apereas almost ready to publish, but minor corrections to the text and figures should be made. The manuscript is clear and well written, but correction of the English language by an English native speaker (if possible) is recommended.

On figure 1D, please correct the label “DI” to “ID” (inflammatory diseases).

On figure 2, several lanes on the western blot appears transfected with control or miRNA-124 lentivirus (why are all samples in duplicates?). Did the authors use different concertation of virus or were all the samples merely repeated twice? Please indicate.

Also, in figure 2, transfection of HH with both control and miRNA-124 appears to have no effect on activation of STAT3 (as indicated by phosphor-STAT3 levels). Could this be explained by reduced transfection by the miRNA-124 lentivirus (diluted by the control virus)? Did the authors analyse the expression levels of miRNA-124 in transfected cells, and if so- could the miRNA-124 level be correlated to effect on phosphor-STAT3?

On figure 3, please indicate what the colour codes (blue/red) means in connection to modulation of gene expression (up/down). This is not clearly indicated on the figure.

Author Response

Reviewer 2

Overall, the manuscript appears almost ready to publish, but minor corrections to the text and figures should be made. The manuscript is clear and well written, but correction of the English language by an English native speaker (if possible) is recommended.

Answer: We thank the reviewer for the positive comment. We have revised the text as suggested.

On figure 1D, please correct the label “DI” to “ID” (inflammatory diseases).

Answer: We have corrected it.

On figure 2, several lanes on the western blot appears transfected with control or miRNA-124 lentivirus (why are all samples in duplicates?). Did the authors use different concertation of virus or were all the samples merely repeated twice? Please indicate.

Answer: The two lanes shown in the western blot are biological replicates. This is now indicated in the figure legend.

Also, in figure 2, transfection of HH with both control and miRNA-124 appears to have no effect on activation of STAT3 (as indicated by phosphor-STAT3 levels). Could this be explained by reduced transfection by the miRNA-124 lentivirus (diluted by the control virus)? Did the authors analyse the expression levels of miRNA-124 in transfected cells, and if so- could the miRNA-124 level be correlated to effect on phosphor-STAT3?

Answer: We apologize for having omitted this fact, and thank the reviewer for the observation. We have now mentioned this possibility it in the figure legend: “Of note, combination of control virus and miRNA-124 in HH did not produce any effect on p-STAT3 likely due to the dilution of miRNA-124 lentivirus”. Unfortunately, we have not tested the levels of miRNA-124 expressed in the transduced cells.

On figure 3, please indicate what the color codes (blue/red) means in connection to modulation of gene expression (up/down). This is not clearly indicated on the figure

Answer: The legend for figure 3 has been changed for clarity to: “Heatmap showing the top differentially expressed genes (up-regulated, in red or down-regulated, in blue) comparing miR-124-transduced with control non-transduced cells.”

Reviewer 3 Report

The manuscript is well-written and paves the way to analyse the STAT3 pathway as a possible therapeutic target.

Author Response

We thank you for the positive evaluation of our work

Reviewer 4 Report

In manuscript, the authors provided convincing evidence that epigenetic dysregulation of miR-124 contributes to the regulation of STAT3 activation in CTCL.

The experimental approach and the results are well presented and the conclusions are supported by the data presented.

Minor comments:

In Figure 4, are there the experimental repeats for the data that was presented? If so please indicate the range of variations and the statistical significance of the differences seen. 

Author Response

Reviewer 4

In manuscript, the authors provided convincing evidence that epigenetic dysregulation of miR-124 contributes to the regulation of STAT3 activation in CTCL.

The experimental approach and the results are well presented and the conclusions are supported by the data presented.

Answer: We thank the reviewer for the positive comment.

Minor comments:

In Figure 4, are there the experimental repeats for the data that was presented? If so, please indicate the range of variations and the statistical significance of the differences seen. 

Answer: The western blot analysis in 4A has been repeated a minimum of 3 times for each cell line and we now mention it in the figure legend. Survival analysis in 4B was done twice and we originally included results from the same experiment shown in 4A. We are now including the average value of the two experiments with deviation and stats.